# Design, Modeling, and Model-Free Control of Permanent Magnet-Assisted Synchronous Reluctance Motor for e-Vehicle Applications

**Songklod Sriprang** [1,2], **Nitchamon Poonnoy** [2,*], **Babak Nahid-Mobarakeh** [3], **Noureddine Takorabet** [1], **Nicu Bizon** [4], **Pongsiri Mungporn** [5] and **Phatiphat Thounthong** [2,*]

1   Groupe de Recherche en Energie Electrique de Nancy (GREEN), Université de Lorraine, GREEN, F-54000 Nancy, France; songklod.sriprang@univ-lorraine.fr (S.S.); noureddine.takorabet@univ-lorraine.fr (N.T.)
2   Renewable Energy Research Centre (RERC), Department of Teacher Training in Electrical Engineering, Faculty of Technical Education, King Mongkut's University of Technology North Bangkok, 1518, Pracharat 1 Road, Bangsue, Bangkok 10800, Thailand
3   Department of Electrical and Computer Engineering, McMaster University, Hamilton, ON L8S 4L8, Canada; babak.nahid@mcmaster.ca
4   Faculty of Electronics, Communications and Computers, University of Pitesti, Arges, 110040 Pitesti, Romania; nicu.bizon@upit.ro
5   Thai-French Innovation Institute (TFII), King Mongkut's University of Technology North Bangkok, 1518, Pracharat 1 Road, Bangsue, Bangkok 10800, Thailand; pongsiri.m@tfii.kmutnb.ac.th
*   Correspondence: nitchamon.p@fte.kmutnb.ac.th (N.P.); phatiphat.t@fte.kmutnb.ac.th (P.T.)

**Abstract:** This paper describes the model-free control approaches for permanent magnet-assisted (PMa) synchronous reluctance motors (SynRMs) drive. The important improvement of the proposed control technique is the ability to determine the behavior of the state-variable system during both fixed-point and transient operations. The mathematical models of PMa-SynRM were firstly written in a straightforward linear model form to show the known and unknown parts. Before, the proposed controller, named here the intelligent proportional-integral (*i*PI), was applied as a control law to fix some unavoidable modeling errors and uncertainties of the motor. Lastly, a dSPACE control platform was used to realize the proposed control algorithm. A prototype 1-kW test bench based on a PMa-SynRM machine was designed and realized in the laboratory to test the studied control approach. The simulation using MATLAB/Simulink and experimental results revealed that the proposed control achieved excellent results under transient operating conditions for the motor drive's cascaded control compared to traditional PI and model-based controls.

**Keywords:** electric vehicle; inverter; permanent magnet-assisted synchronous reluctance motor; PMa-SynRM; model-free control; traction drive

## 1. Introduction

By the end of 2021, the demand for electrical traction machines, including battery electric vehicles and hybrid electric vehicles (HEVs), surpassed two million units [1–4]. Electrical traction machines are also required to further develop more electric aircrafts (MEAs) [5–7]. For these reasons, several state-of-the-art machines have been developed in the last few years, such as synchronous reluctance motors (SynRMs) and especially permanent magnet-assisted synchronous reluctance motors (PMa-SynRMs). PMa-SynRMs can produce 75% of the torque of an interior permanent magnet synchronous motor (IPMSM) for the same size and liquid cooling technology [8,9]. In addition, state-of-the-art modern motors provide more desired characteristics for electric vehicle (EV) applications, in particular, high efficiency at low and high speeds. Therefore, PMa-SynRMs constitute a promising choice for these applications. However, PMa-SynRMs have a much more complicated structure, which affects the control system, and its model is strongly nonlinear. Therefore,

traditional control, such as field-oriented control (FOC) based on a proportional-integral (PI) controller, cannot accomplish high performance for all operating conditions of these modern machines.

Furthermore, in EV applications, safety, energy saving, and soft driving are mandatory and require improvement of the control performance of the motor drive system. Many studies have been conducted in the last few years regarding SynRMs and PMa-SynRMs, with special attention to machine design and optimization aspects. Multiple-flux barrier rotors and transversely laminated rotors were reported. Rotor laminations are made by traditional punching or wire cutting, resulting in easy and cheap construction [10,11]. Control characteristics have also been investigated [9,12,13]. In this regard, in the current control of PMa-SynRM drive systems, the essential objective is to ensure that the stator currents track the reference values with minimum errors in both transient and steady-state conditions. To design a robust controller with acceptable tracking performance, all the model-based control (MBC) approaches mentioned in the literature applied to PMa-SynRM require extensive knowledge about the dynamics and the model of PMa-SynRM systems. In addition, the MBC performance can be affected by unexpected dynamic variations of the system and parametric uncertainty, which are very common phenomena in industrial applications. To overcome the limitations of MBC approaches, some studies proposed model predictive control (MPC) as an appropriate current control scheme for electric motors, which ensures a fast dynamic and a remarkable safety factor [14–16]. This method's concept is based on predicting controlled variables in the next calculation step using the measured variables and a mathematical model of the controlled system. Then, the predicted results are analyzed using a cost function in terms of the difference between the desired trajectories and real outputs of the system. Compared to the previously mentioned control techniques, safety and fast dynamics are two remarkable features of the MPC method. Despite these advantages, the performance of MPC highly depends on the correctness of the model, given that a mathematical model is used in the prediction section [17]. When using the prediction at each sampling time of the MPC algorithm, some additional mathematical calculations are imported into the control algorithm.

Therefore, a control principle called model-free control (MFC) has been proposed to address the limitations of the abovementioned MPC and MBC techniques. MFC, also referred to as model-free tuning in the literature, uses a local linear approximation of the process model, which is valid for a small time window, and a fast estimator, which is employed to update the approximation [18,19]. The main advantage of MFC is that it does not require the process model in the controller tuning. Few experiments have been conducted on real-world control system structures concerning the tuning process. This paper introduces MFC development to control both torque and speed control of PMa-SynRMs. To verify the advantages of MFC, both simulations and experiments were carried out under several conditions.

This paper is organized as follows. A model-free control and control law are briefly introduced in Section 2. The main issues regarding the control of PMa-SynRMs, related state-of-the-art studies, as well as mathematical models are reviewed in Section 3, with a focus on MFC applied to PMa-SynRM drive systems. In Section 4, simulation and experimental results are provided to demonstrate the advantages of the proposed MFC. Sections 5 and 6 summarize and conclude the paper. A small-scale 1-kW test bench based on a PMa-SynRM with ferrite magnets was implemented to confirm the high performance of the designed control scheme in the laboratory [13].

## 2. Model-Free Control and Control Law (Brief Introduction)

### 2.1. Model-Free Control

The idea of model-free control accomplished for control system applications was originally proposed by Fliess et al. [20,21]. Many industrial applications have significantly changed with technology development and have become more complex. Accordingly, modeling the dynamic and process of these applications using mathematical models

becomes very difficult or at least time-consuming. In this case, using the MBC methods for these kinds of applications will be impossible. Conversely, almost all industrial applications generate and save a large number of process data that contain all the necessary information related to the system's operation. In this case, it is important to use these generated data, obtained online/offline, directly for designing the controller or other purposes. In this way, the model-free control (MFC) foundation is essential in controlling industrial applications. So far, the types of the modern control system can be roughly categorized by MBC and MFC, as in Figure 1.

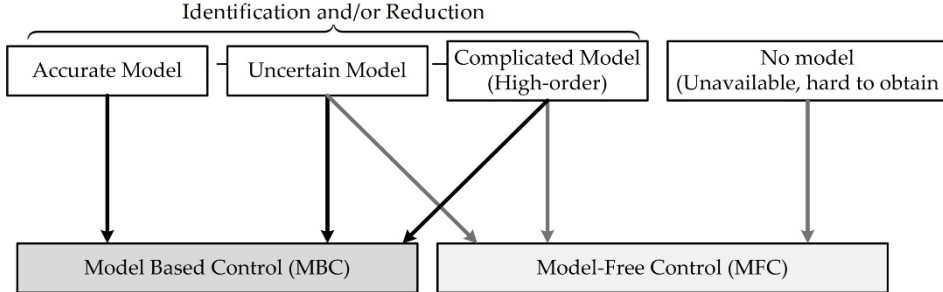

**Figure 1.** Control law's block diagram.

MFC is a control method that uses only the online data obtained from the controlled system to design the controller, without the additional need for information about the mathematical model or parameters of the studied system. Therefore, the MFC can be applicable for all nonlinear systems with complex or unknown structures.

The principle of model-free control is briefly introduced next. A nonlinear system can be described by a state-variable written as follows:

$$\dot{x} = f(x, u)$$
$$y = h(x, u) \tag{1}$$

where

$$x = [x_1, x_2, \ldots, x_n]^T; x \in \mathbb{R}^n$$
$$u = [u_1, u_2, \ldots, u_m]^T; u \in \mathbb{R}^m$$
$$y = [y_1, y_2, \ldots, y_m]^T; y \in \mathbb{R}^m \tag{2}$$

where $x$ is the state variable, $u$ is the control variable, $y$ is the output variable, and $n$, $m \in$ N.

According to Equation (2), the system described by Equation (1) is flat. A control law of variable $u$ can be expressed as follows [13]:

$$u = u_{\text{ref}} + u_{\text{feedback}}(\varepsilon) \tag{3}$$

with $\varepsilon = y_{\text{ref}} - y$.

This control law is suitable for all systems with known parameters. However, if only some system parameters can be identified or the system described by Equation (1) cannot be identified, the controller needs to be modified as a partially-known model, replaced by a model-free control as follows:

$$u = \frac{\hat{\alpha}(y, \dot{y}, \ddot{y}, \ldots, y^{(n)})}{b} + \frac{F}{b} \tag{4}$$

where $\hat{\alpha}(\dot{y})/b$ is a known system, and $F$ denotes an unknown part of the system.

The difference between $\hat{\alpha}(y, \dot{y}, \ddot{y}, \ldots, y^{(n)})$, $\hat{\alpha}(\dot{y})$, and $\dot{y}$ is that the $\hat{\alpha}(y, \dot{y}, \ddot{y}, \ldots, y^{(n)})$ is the known part of the $\alpha(y, \dot{y}, \ddot{y}, \ldots, y^{(k)})$, the $\hat{\alpha}(\dot{y})$ is the only known part of the studied system, and the $\dot{y}$ is the differential of the known part, respectively.

Alternatively, it can be rewritten and rearranged as a straightforward linear model as follows:

$$\dot{y} = -F + b \cdot u \tag{5}$$

## 2.2. Control Law

Figure 2 represents the control law block diagram for the model-free control technique. The control law is defined as follows:

$$u = u_{\text{ref}} + u_{\text{feedback}}(\varepsilon) + \frac{\widehat{F}}{b} \tag{6}$$

where

$$u_{\text{ref}} = \frac{\hat{\alpha}\left(y_{\text{ref}}, \dot{y}_{\text{ref}}, \ddot{y}_{\text{ref}}, \cdots y_{\text{ref}}^{(\beta+1)}\right)}{b} \tag{7}$$

and $\widehat{F}$ is the estimated value of $F$, which is expressed as follows:

$$\widehat{F} = b \cdot u - \dot{y} \tag{8}$$

The function $\hat{\alpha}$ is a regular function [22,23].

The feedback term $u_{\text{feedback}}$ can be described by applying the PI controller as follows:

$$u_{\text{feedback}} = K_{\text{p}} \cdot \varepsilon + K_{\text{i}} \cdot \int \varepsilon dt \tag{9}$$

Substituting Equation (6) into Equation (5), and rearranging the expressions, Equation (5) can be expressed as follows:

$$\dot{y} = -F + b \cdot u_{\text{ref}} + b \cdot u_{\text{feedback}}(\varepsilon) + \widehat{F} \tag{10}$$

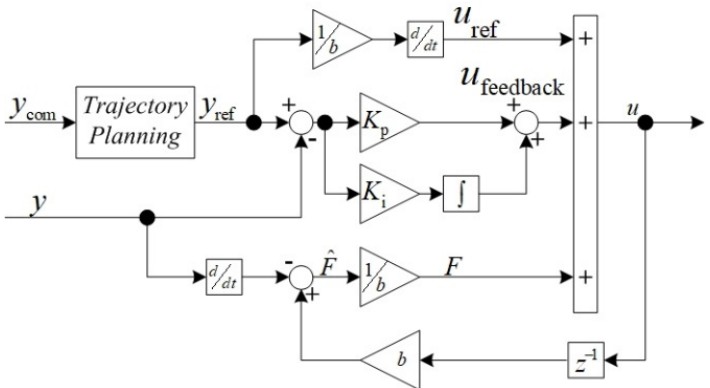

**Figure 2.** Control law's block diagram.

## 2.3. Controller Design

The observation term purposes to afford an estimated signal $\widehat{F}$ so that $\widehat{F} \to F$ as $t \to \infty$ (under global convergence assumption for the estimation). Consequently, Equation (10) can be rewritten as follows:

$$\dot{y} = b \cdot u_{\text{ref}} + b \cdot u_{\text{feedback}}(\varepsilon) \tag{11}$$

Consequently, Equation (11) describes the dynamic of the closed-loop control system. By substituting Equation (9) into Equation (11) and rearranging, Equation (11) can be expressed as follows:

$$\frac{d(y_{ref} - y)}{dt} + b \cdot K_p \cdot \varepsilon + b \cdot K_i \int \varepsilon dt = 0 \tag{12}$$

Referring to the control law displayed in Figure 2, the controller coefficients can be determined using the following expression obtained by taking time derivation in Equation (12):

$$\ddot{\varepsilon} + b \cdot K_p \cdot \dot{\varepsilon} + b \cdot K_i \cdot \varepsilon = 0 \tag{13}$$

Comparing Equation (13) to the 2nd order standard equation stated as follows:

$$\ddot{q} + 2 \cdot \zeta \cdot \omega_n \cdot \dot{q} + \omega_n^2 \cdot q = 0 \tag{14}$$

the controller coefficients become:

$$K_p = \frac{2 \cdot \zeta \cdot \omega_n}{b} \tag{15}$$

and

$$K_i = \frac{\omega_n^2}{b} \tag{16}$$

where $\zeta$ and $\omega_n$ are the tuning dominant damping ratio and natural frequency, respectively.

The gain $b \in \mathbb{R}$ is a non-physical constant parameter. Instead of $\alpha$, the $b$ is present in this paper, as shown in (4). It was chosen by the practitioner or obtained by trials and errors. *F*, which is continuously updated, subsumes the poor parts of the plant and the various possible disturbances without distinguishing between them [24,25]

### 3. Applying Model-Free Control to PMa-SynRM Drive

*3.1. Mathematic Model of PMa-SynRM/Inverter*

A variable speed drive (VSD), which powers the PMa-SynRM under study, is shown in Figure 3. Owing to the rotor geometries of the PMa-SynRM discussed in [26], the current control strategies in the literature differ from those applied to PMSM. The rotor geometries of PMa-SynRMs are given by the salient-pole, in which $L_d > L_q$. Its torque expression was given by Equation (17). In this case, the $i_d$ component should not be equal to zero to take advantage of the reluctance torque produced by the high saliency ratio. Therefore, the maximum torque per ampere (MTPA) control strategy was recommended for PMa-SynRMs. The main idea of this control was to develop the requested torque using the minimum value of the stator current magnitude:

$$T_e = n_p\{\Psi_m - (L_d - L_q)i_q\} \cdot i_d \tag{17}$$

The equations of a PMa-SynRM in the rotating $d_q$ reference frame and a mechanical equation are expressed by a state-space representation as follows:

$$\underbrace{\begin{bmatrix} \frac{di_d}{dt} \\ \frac{di_q}{dt} \\ \frac{d\omega_m}{dt} \end{bmatrix}}_{\dot{x}} = \underbrace{\begin{bmatrix} \{-R_s i_d + \omega_e(L_q i_q - \Psi_m)\}/L_d \\ (-R_s i_q - \omega_e L_d i_d)/L_q \\ [n_p\{\Psi_m i_d + (L_d - L_q)i_q i_d\} - B_f \omega_m]/J \end{bmatrix}}_{f(x)} + \underbrace{\begin{bmatrix} \frac{1}{L_d} & 0 & 0 \\ 0 & \frac{1}{L_q} & 0 \\ 0 & 0 & -\frac{1}{J} \end{bmatrix}}_{\mathbf{B}} \underbrace{\begin{bmatrix} v_d \\ v_q \\ T_L \end{bmatrix}}_{u}$$

$$y = \underbrace{\begin{bmatrix} 1 & 0 & 0 \\ 0 & 1 & 0 \\ 0 & 0 & 1 \end{bmatrix}}_{\mathbf{C}} \begin{bmatrix} i_d \\ i_q \\ \omega_m \end{bmatrix} \tag{18}$$

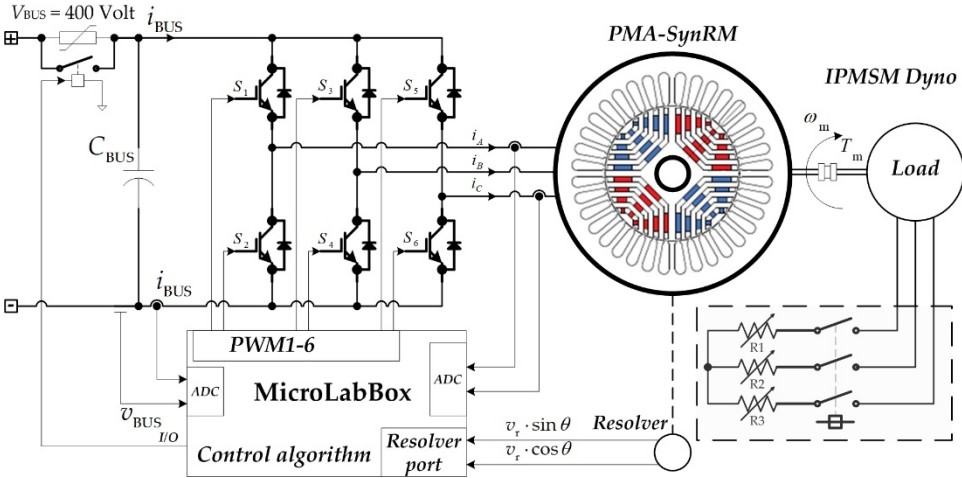

**Figure 3.** A three-phase inverter to control a PMa-SynRM prototype.

### 3.2. Model-Free of Current and Speed Control Development

The control system of PMa-SynRMs proposed in this paper (Figure 4) had a case cascade construction consisting of two loops (i.e., inner current control loop and outer speed control loop). The inner current loop was much faster than the outer speed control loop, such that the model-free control for the current control was developed first. By defining $u = [u_1 \ u_2]^T = [v_d \ v_q]^T$, $y = [y_1 \ y_2]^T = [i_d \ i_q]^T$, and rearranging the first and second rows in Equation (18) in the form of Equation (5), the PMa-SynRM model is expressed as follows:

$$\frac{di_d}{dt} = -\frac{R_s i_d}{L_d} + \frac{\omega_e(L_q i_q - \Psi_m)}{L_d} + v_d \cdot \frac{1}{L_d}$$
$$\frac{di_q}{dt} = -\frac{R_s i_q}{L_q} - \frac{\omega_e L_d i_d}{L_q} + v_q \cdot \frac{1}{L_q} \tag{19}$$

According to the principle of the model-free as in [20,21], Equation (19) can be separated to identify the known and unknown terms as follows. The known terms are

$$\hat{\alpha}_1 = \frac{\dot{y}_1}{b_1} = L_d \frac{di_d}{dt}$$
$$\hat{\alpha}_1 = \frac{\dot{y}_2}{b_2} = L_q \frac{di_q}{dt} \tag{20}$$

and the unknown terms are

$$F_1 = \left\{ -R_s i_d + \omega_e(L_q i_q - \Psi_m) \right\} \cdot \frac{1}{L_d}$$
$$F_2 = \left( -R_s i_q - \omega_e L_d i_d \right) \cdot \frac{1}{L_q} \tag{21}$$

According to the control law (Figure 2), the first term of the model-free control for inner current loop control is determined as follows:

$$u_{1ref} = \frac{\dot{y}_{1ref}}{b_1} = L_d \frac{di_d}{dt}$$
$$u_{2ref} = \frac{\dot{y}_{2ref}}{b_2} = L_q \frac{di_q}{dt} \tag{22}$$

The estimation of unknown terms is expressed as follows:

$$\widehat{F}_1 = \frac{1}{L_d} u_1 - \dot{y}_1 = \frac{1}{L_d} v_d - \frac{di_d}{dt}$$
$$\widehat{F}_2 = \frac{1}{L_q} u_1 - \dot{y}_2 = \frac{1}{L_q} v_q - \frac{di_q}{dt} \tag{23}$$

The feedback terms of *d*- and *q*-axis current control are obtained as follows:

$$b_1 \cdot u_{1\text{feedback}} = b_1 \left( K_{\text{pd}} \cdot \varepsilon_{\text{d}} + K_{\text{id}} \int \varepsilon_{\text{d}} dt \right)$$
$$b_2 \cdot u_{2\text{feedback}} = b_2 \left( K_{\text{pq}} \cdot \varepsilon_{\text{q}} + K_{\text{iq}} \int \varepsilon_{\text{q}} dt \right) \tag{24}$$

Concerning the design procedure in the controller design, Equation (24) can be rewritten as follows:

$$\ddot{\varepsilon}_{\text{d}} + b_1 \cdot K_{\text{pd}} \cdot \dot{\varepsilon}_{\text{d}} + b_1 \cdot K_{\text{id}} \cdot \varepsilon_{\text{d}} = 0$$
$$\ddot{\varepsilon}_{\text{q}} + b_2 \cdot K_{\text{pq}} \cdot \dot{\varepsilon}_{\text{q}} + b_2 \cdot K_{\text{iq}} \cdot \varepsilon_{\text{d}} = 0 \tag{25}$$

The controller coefficients $K_{\text{pd}}$, $K_{\text{id}}$, $K_{\text{pq}}$, and $K_{\text{iq}}$ are determined as follows:

$$K_{\text{pd}} = \frac{2\zeta_1 \omega_{\text{n1}}}{b_1}, K_{\text{id}} = \frac{\omega_{\text{n1}}^2}{b_1}$$
$$K_{\text{pq}} = \frac{2\zeta_1 \omega_{\text{n1}}}{b_2}, K_{\text{iq}} = \frac{\omega_{\text{n1}}^2}{b_2} \tag{26}$$

The second model-free control for the outer speed control loop is developed here. The output of the speed control loop provides the torque reference of the MTPA algorithm, generating optimized *d*- and *q*-axis current references. Therefore, $T_{\text{e}}$ was chosen as a control variable of the outer speed control loop, such that $u_3 = T_{\text{eREF}}$. Then, rewriting the mechanical equation of the PMa-SynRM represented by the third row in Equation (18) in the form of Equation (5) yields:

$$\frac{d\omega_{\text{m}}}{dt} = \left( -B_f \cdot \omega_{\text{m}} - T_{\text{L}} \right) \cdot \frac{1}{J} + T_{\text{e}} \cdot \frac{1}{J} \tag{27}$$

Separating this equation into the known and unknown terms, the known term is expressed as follows:

$$\hat{\alpha}_3 = \frac{\dot{y}_3}{b_3} = J \frac{d\omega_{\text{m}}}{dt} \tag{28}$$

The unknown term is expressed as follows:

$$F_3 = \left( -B_f \omega_m - T_L \right) \cdot \frac{K_t}{J} \tag{29}$$

Each part of the model-free control for the outer speed control loop is defined according to the following expression:

$$u_{3\text{ref}} = \frac{\dot{y}_{3\text{ref}}}{b_3} = J \frac{d\omega_{\text{m}}}{dt} \tag{30}$$

The estimation of the unknown term is expressed as follows:

$$\widehat{F}_3 = \frac{K_t}{J} u_3 - \dot{y}_3 = \frac{1}{J} \cdot T_{\text{e}} - \frac{d\omega_{\text{m}}}{dt} \tag{31}$$

$$b_3 u_{3\text{feedback}} = b_3 \left( K_{\text{p}\omega} \cdot \varepsilon_{\omega} + K_{\text{i}\omega} \int \varepsilon_{\omega} dt \right) \tag{32}$$

Regarding the controller design procedure, Equation (32) can be rewritten as follows:

$$\ddot{\varepsilon}_{\omega} + b_1 \cdot K_{\text{p}\omega} \cdot \dot{\varepsilon}_{\omega} + b_1 \cdot K_{\text{i}\omega} \cdot \varepsilon_{\omega} = 0 \tag{33}$$

The controller coefficients $K_{\text{p}\omega}$ and $K_{\text{i}\omega}$ are determined as follows:

$$K_{\text{p}\omega} = \frac{2\zeta_2 \omega_{\text{n2}}}{b_3}, K_{\text{i}\omega} = \frac{\omega_{\text{n2}}^2}{b_3} \tag{34}$$

where $\zeta_2$ and $\omega_{n2}$ are the desired dominant damping ratio and natural frequency of the outer speed control loop, respectively.

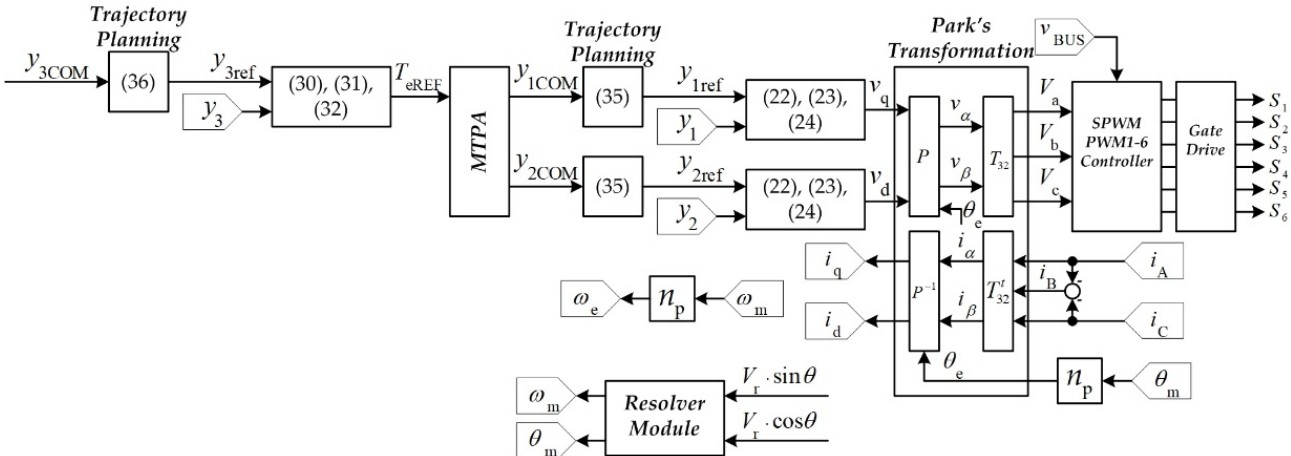

**Figure 4.** Control system of PMa-SynRM based on a model-free control diagram.

### 3.3. Trajectory Planning

Finally, as presented in Figure 2, desired trajectory planning must be implemented to generate the input set-point $y_{REF}$. A second order filter is often implemented to plan the desired trajectory for the controlled output. It permits limiting the derivative terms in the control law. The proposed trajectory planning for the two inner current control loops is expressed as follows:

$$\frac{y_{1REF}}{y_{1COM}} = 1 \bigg/ \left\{ \left( \frac{s}{\omega_{n3}} \right)^2 + \frac{2\zeta_3}{\omega_{n3}} s + 1 \right\} \tag{35}$$

$$\frac{y_{2REF}}{y_{2COM}} = 1 \bigg/ \left\{ \left( \frac{s}{\omega_{n3}} \right)^2 + \frac{2\zeta_3}{\omega_{n3}} s + 1 \right\} \tag{36}$$

where $\zeta_3$ and $\omega_{n3}$ are the tuning dominant damping ratio and natural frequency, respectively.

The trajectory planning of the outer speed loop is expressed as follows:

$$\frac{y_{3REF}}{y_{3COM}} = 1 \bigg/ \left\{ \left( \frac{s}{\omega_{n4}} \right)^2 + \frac{2\zeta_4}{\omega_{n4}} s + 1 \right\} \tag{37}$$

where $\zeta_4$ and $\omega_{n4}$ are the desired dominant damping ratio and natural frequency of the speed loop trajectory planning, respectively.

## 4. Simulation and Experimental Validation of the Model-Free Control Applied to PMa-SynRM

### 4.1. Experimental Setup

A small-scale test bench 1-KW relying on the prototype PMa-SynRM was conceived in the laboratory, as shown in Figure 5. The prototype PMa-SynRM was supplied by a 3-kW 3-phase inverter (DC/AC) operating at a switching frequency of 16 kHz. Besides, the input DC grid voltage of the inverter was fed by a three-phase variable power supply combined with a three-phase diode rectifier. The PMa-SynRM was mechanically coupled with an IPMSM (interior permanent magnet synchronous motor) feeding a resistive load (see Figure 3). The measurements for the speed and rotor angle were acquired by a resolver placed on the rotor shaft. The developed control scheme relying on the model-free control was modeled in the Matlab/Simulink software, and then it was incorporated in the dSPACE 1202 MicroLabBox real-time interface to generate the gate control signals applied to the VSI.

The main PMa-SynRM parameters are listed in Table 1, whereas the model-free controller parameters are listed in Table 2.

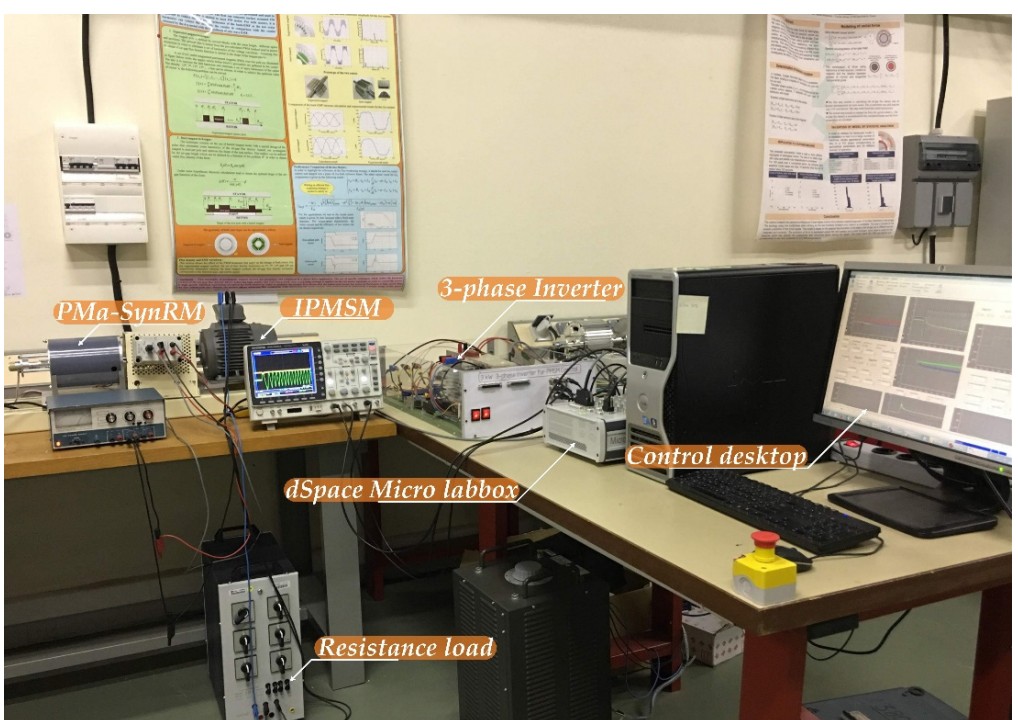

**Figure 5.** Experimental setup.

**Table 1.** Specifications and parameters of the motor/inverter.

| Symbol | Quantity | Value |
|---|---|---|
| $P_{\text{rated}}$ | Rated power | 1 kW |
| $n_{\text{rated}}$ | Rated speed | 1350 rpm |
| $T_{\text{rated}}$ | Rated torque | 7.07 Nm |
| $n_{\text{p}}$ | Number of pole pairs | 2 |
| P.F. | Power factor | 0.80 |
| $R_{\text{s}}$ | Resistance (motor + inverter) | 3.2 Ω |
| $L_{\text{d}}$ | Nominal $d$-axis inductance | 288 mH |
| $L_{\text{q}}$ | Nominal $q$-axis inductance | 38 mH |
| $J$ | Equivalent inertia | 0.017 kg m$^2$ |
| $B_{\text{f}}$ | Viscous friction coefficient | 0.008 Nm s/rad |
| $\Psi_{\text{m}}$ | PMs flux linkage | 0.138 Wb |
| $f_{\text{s}}$ | Switching frequency | 16 kHz |
| $V_{\text{dc}}$ | DC bus voltage | 400 V |

**Table 2.** Current/torque and speed regulation parameters.

| Symbol | Quantity | Value |
|---|---|---|
| $\zeta_{1\text{d}}$ | Damping ratio 1 | 0.7 |
| $\omega_{\text{n1d}}$ | Natural frequency 1 | 3000 Rad s$^{-1}$ |
| $\zeta_{1\text{q}}$ | Damping ratio 1 | 0.7 pu. |
| $\omega_{\text{n1q}}$ | Natural frequency 1 | 2000 Rad s$^{-1}$ |
| $\zeta_2$ | Damping ratio 2 | 0.7 |

**Table 2.** *Cont.*

| Symbol | Quantity | Value |
|---|---|---|
| $\omega_{n2}$ | Natural frequency 2 | 107.1419 Rad s$^{-1}$ |
| $\zeta_{3d}$ | Damping ratio 3 | 1 |
| $\omega_{n3d}$ | Natural frequency 3 | 300 Rad s$^{-1}$ |
| $\zeta_{3q}$ | Damping ratio 3 | 1 |
| $\omega_{n3q}$ | Natural frequency 3 | 200 Rad s$^{-1}$ |
| $\zeta_4$ | Damping ratio 4 | 1 |
| $\omega_{n4}$ | Natural frequency 4 | 150 Rad s$^{-1}$ |
| $T_{emax}$ | Maximum torque reference | +6 Nm |
| $T_{emin}$ | Minimum torque reference | −6 Nm |
| Vdc | DC bus voltage | 400 V |
| $f_s$ | Switching frequency | 16 kHz |

### 4.2. Simulations

The developed MFC algorithm for the PMa-SynRM drive was simulated under different operation conditions before its implementation. Figure 6 shows the simulation results of the set-point tracking *d*-axis inner loop current control response using the model-free control. Interestingly, note that, during the transient response, the *d*-axis current tracked the reference very well, and there was no steady-state error. The simulation conditions were set as follows: for *d*-axis testing, the *q*-axis current command $i_{qCOM}$ was set to zero.

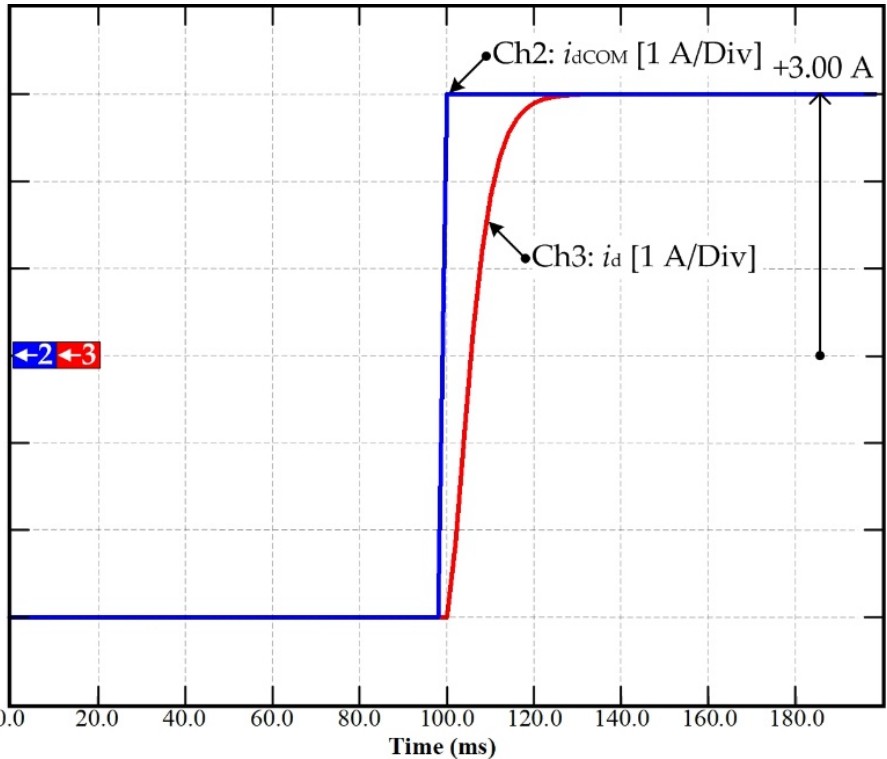

**Figure 6.** Simulation results: Dynamic response of the set-point tracking *d*-axis current control with the MFC.

Figure 7 shows the set-point tracking *q*-axis current control simulation results using the model-free control. Note that the control performance was satisfactory, with good set-point tracking and zero steady-state error. The simulation conditions were set as follows: during *q*-axis testing, the *d*-axis command $i_{dCOM}$ was set to zero, and the load was the rated one.

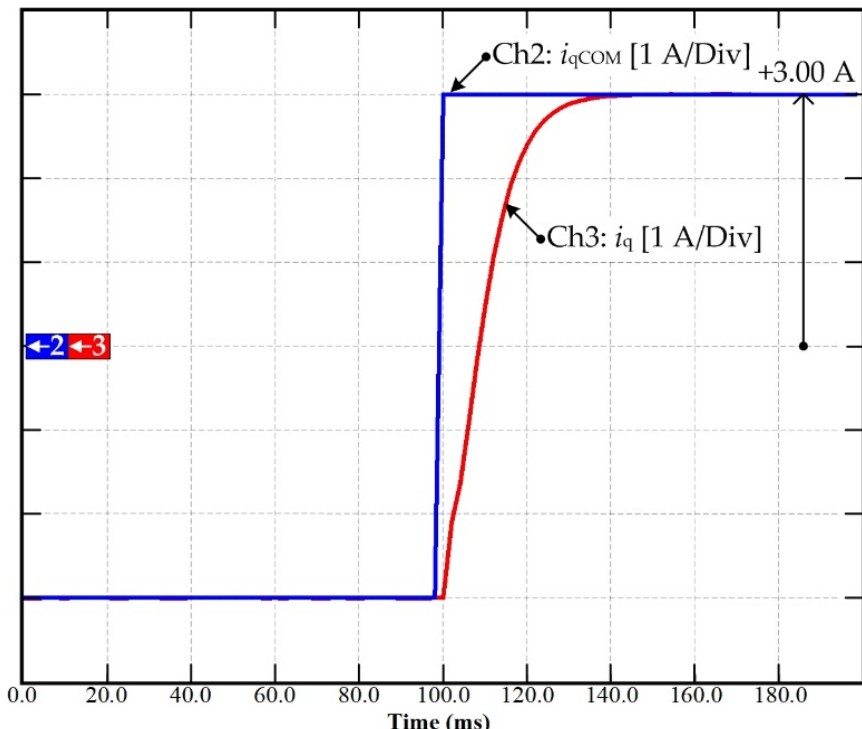

**Figure 7.** Simulation results: Dynamic response of $q$-axis currents with the MFC applied to the PMa-SynRM drive.

Another simulation result is depicted in Figure 8. It shows the drive response to a step change on the speed reference from 0 to 1000 rpm. In this figure, Chs 1, 2, and 4 represent the speed command $n_{COM}$, speed reference $n_{REF}$, and measured speed $n$, respectively. Chs 3, 5, and 6 represent the torque reference $T_{eREF}$ and the $d$- and $q$-axis currents $i_d$ and $i_q$, and Chs 7 and 8 represent the $d$-axis voltage and $q$-axis voltage, respectively. The parameters of the simulated drive are those of the test bench that will be later used for experimental validation. They are reported in Section 4.1. The MFC was designed to keep the torque within the range $\pm 6$ Nm. Note that the speed response was satisfactory with small overshoot and without steady-state error.

Although no torque sensor was employed in the experimental setup, the torque seemed to be limited to the allowed range. Moreover, $i_q$ and $i_d$ were generated on the basis of the MTPA algorithm discussed in [22].

Figure 9 shows the simulation of the disturbance rejection ability of the MFC applied to the PMa-SynRM drive. In this figure, Ch 4 represents the measured speed $n$, Ch2 represents the $d$-axis current $i_d$, Ch3 represents the $q$-axis current $i_q$, and Ch1 represents the torque reference $T_{eREF}$. The simulation conditions were as follows: $n$ = 1000 rpm; sudden increase of 3.7 Nm on the load torque $T_L$ at 0.3 s; and subsequent clearance of the load torque at 0.7 s. Note that, under the action of the proposed model-free control, when the load changed suddenly, the motor speed deviated slightly from its set-point, but it recovered very quickly. Figure 9 also shows the disturbance rejection capability of the MFC. As a result, the speed control performance was significantly improved, confirming the feasibility of the proposed MFC for this application.

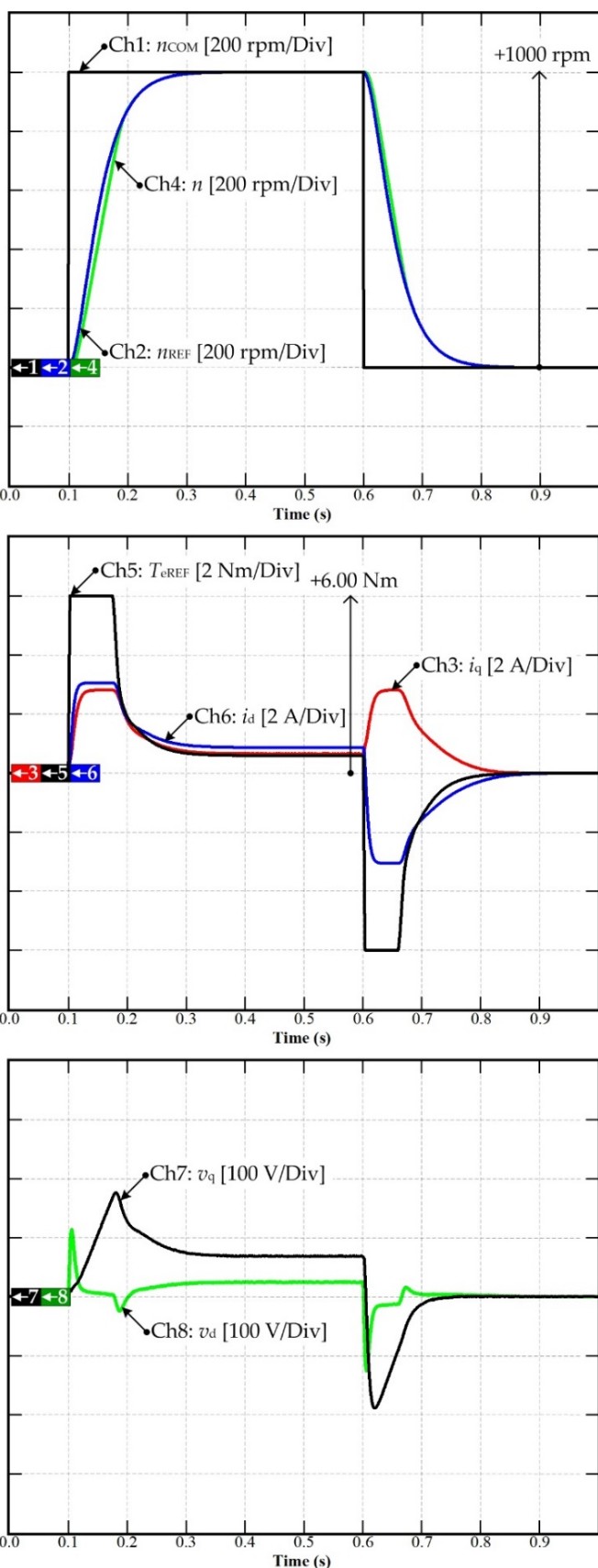

**Figure 8.** Simulation results: Simulated drive response to a 0–1000 rpm reference speed pulse. From top to bottom: speed reference and response, *d*- and *q*-currents, *d*- and *q*-voltages, and active region number of the MFC controller.

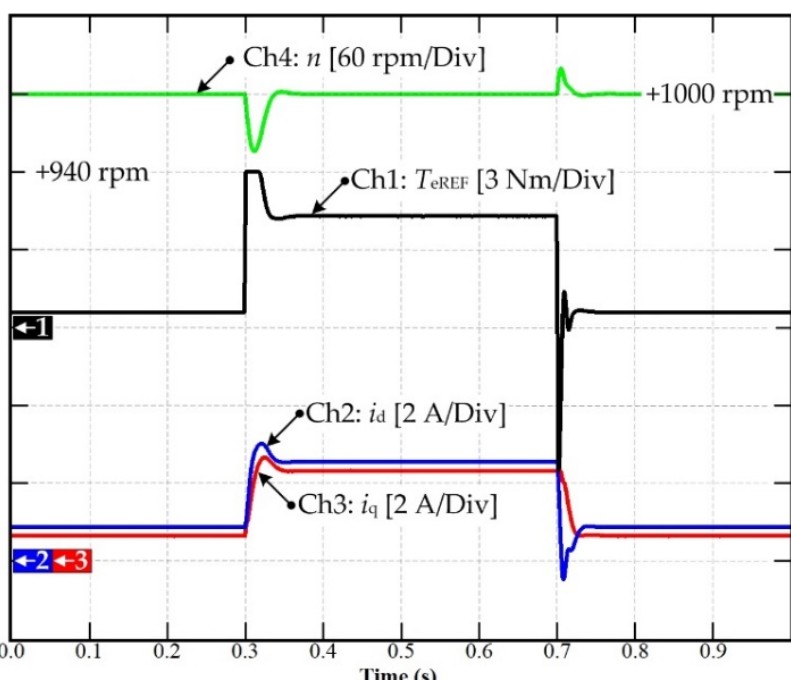

**Figure 9.** Simulation results: Disturbance rejection of MFC applied to the PMa-SynRM drive.

*4.3. Experimental Validation of PMa-SynRM Drive Based on Model-Free Control*

The designed MFC for the PMa-SynRM drive was experimentally validated on a laboratory test bench. The experimental setup is depicted in Figure 5. The entire controller parameters of the current/torque and speed are presented in Table 2. The model-free control stability and response were easy to set compared to the FOC with PI controller. Thus, by defining and selecting the governing damping and natural frequency as mentioned in the literature [19], the controller coefficients of the PI controller for both the current and speed loops control may be calculated by (26) and (34). The PI controller was provided to deal with inevitable modeling errors and uncertainties. Therefore, the PI controller guaranteed the stability of the model-free control the ensure that the current and speed control achieved the steady-state error.

Figure 10 shows the current control test of the set-point tracking *d*-axis inner loop. In this figure, *d*-axis command $i_{dCOM}$, *d*-axis reference $i_{dREF}$, which is provided by the *d*-axis trajectory planning, and the actual *d*-axis current are represented. Ch5, Ch6, and Ch7 represent the measured stator phase currents A, B, and C, respectively. These results are similar to those obtained by simulation and confirm that the current control performance was satisfactory.

The same test was conducted with the *q*-axis current while the *d*-axis current was regulated to zero. In this case, the motor was at a stand-still. Figure 11 depicts the experimental data, where Ch1 represents the *q*-axis current command $i_{qCOM}$, Ch2 represents the *q*-axis current reference $i_{qREF}$, and Ch3 represents the *q*-axis current measurement $i_q$. Ch5, Ch6, and Ch7 represent the measured stator phase currents A, B, and C, respectively. Overall, the current control performance seemed satisfactory.

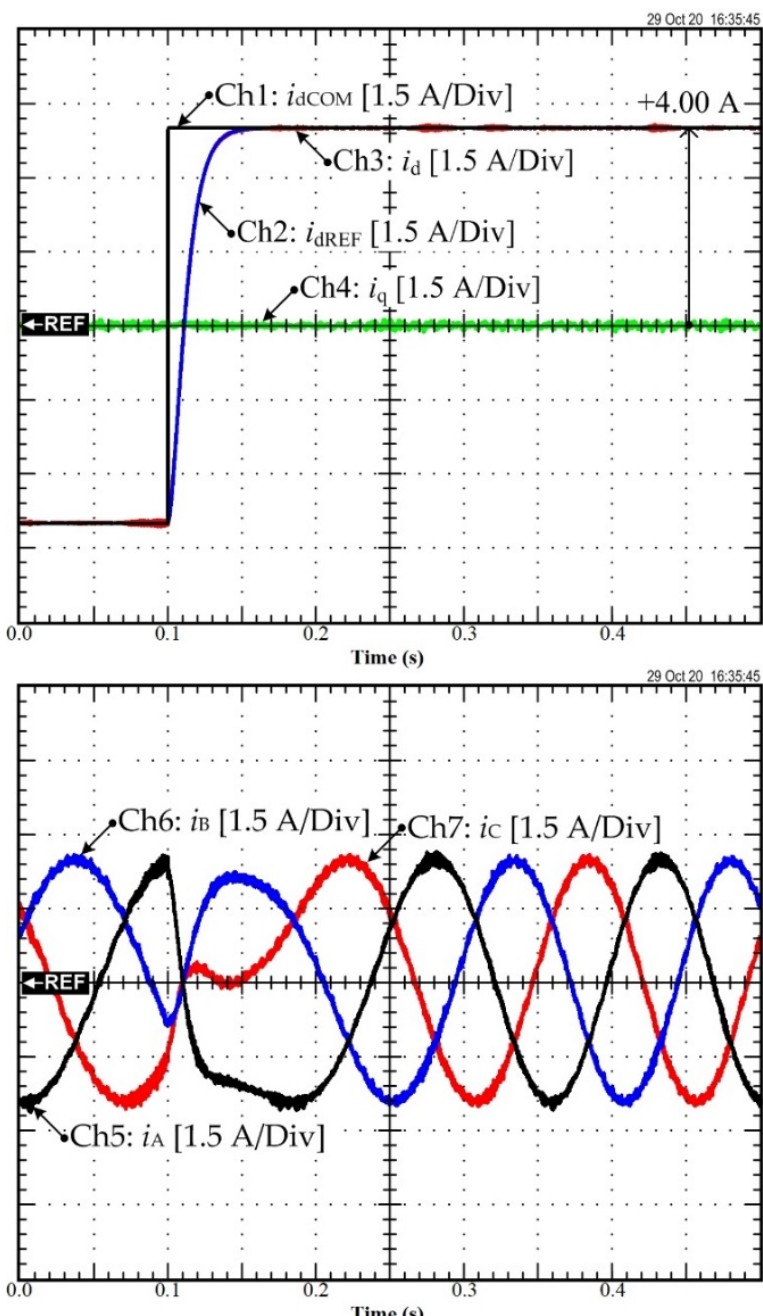

**Figure 10.** Experimental result: Set-point tracking *d*-axis current control response curve based on MFC.

Figure 12 depicts the speed startup of the PMa-SynRM drive using the MFC. In this figure, Chs 1, 2, 3, and 4 represent the torque reference $T_{eREF}$, *d*-axis current $i_d$, *q*-axis current $i_q$, and measured speed *n*, respectively. Chs 5 and 6 represent the output $v_q$ and $v_d$, chosen as the output of the MFC. Moreover, Chs 7 and 8 represent the estimated unknown terms of the *d*- and *q*-axis models. As expected, the torque was limited, and the speed response showed neither overshoot nor steady-state error. It is worth recalling that the torque reference generated $i_q$ and $i_d$ command references according to the MTPA algorithm.

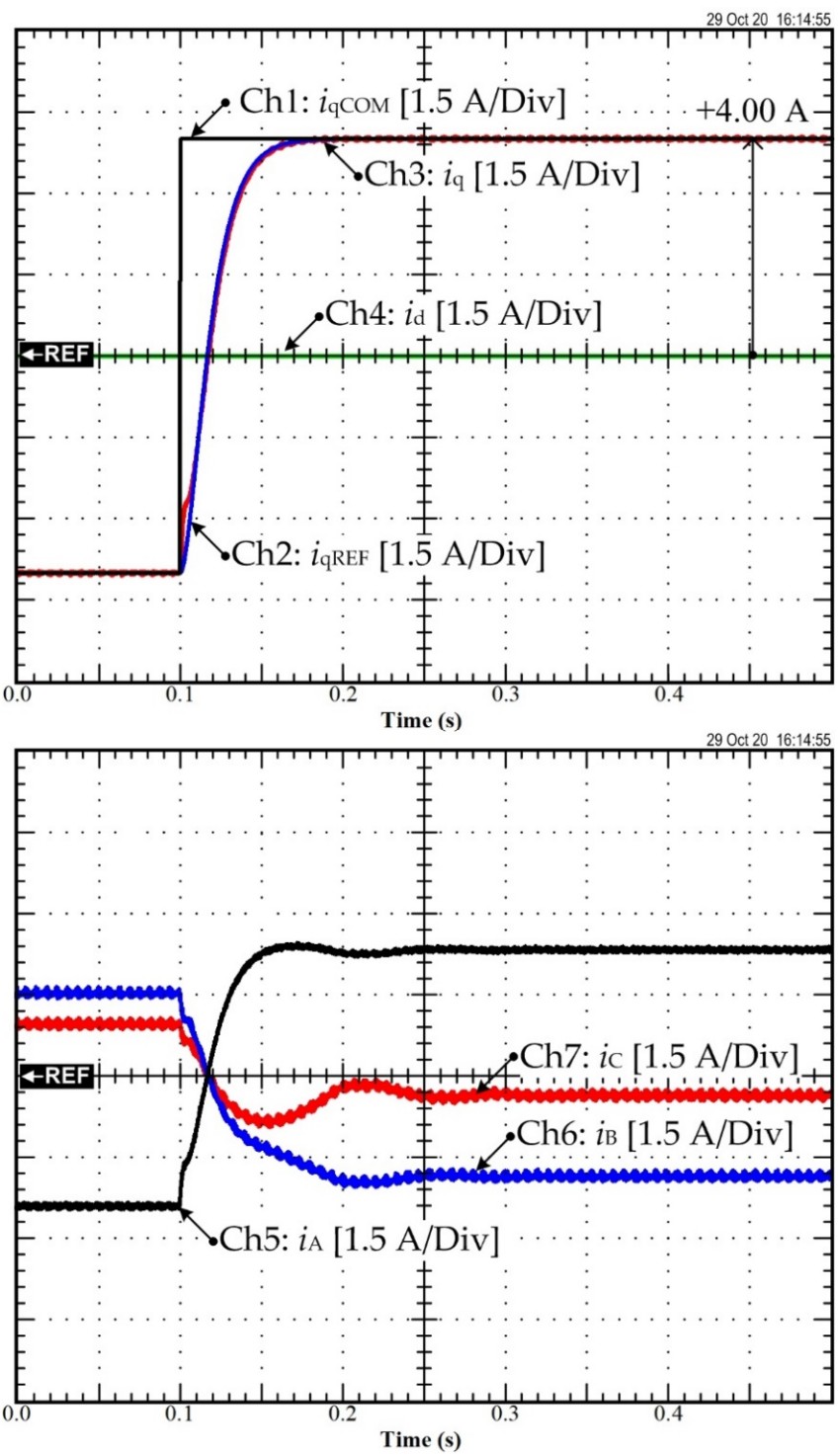

**Figure 11.** Experimental result: Set-point tracking *q*-axis current control response curve based on MFC.

Figure 13 shows the experimental validation of the disturbance rejection ability of the proposed MFC applied to the PMa-SynRM drive. In this figure, Chs 1, 2, 3, and 4 represent the torque reference $T_{eREF}$, *d*-axis current $i_d$, *q*-axis current $i_q$, and measured speed *n*, respectively. Chs 5 and 6 represent the output $v_q$ and $v_d$, chosen as the output of the MFC. Moreover, Chs 7 and 8 represent the estimated unknown terms of the *d*- and *q*-axis current models. The experimental conditions were set as follows: $n_{REF} = 1000$ rpm, and sudden increase of the load torque ($T_L$) to 3.7 Nm at 0.2 s. Note that the proposed

model-free control compensated for the load torque variation and rejected its effect on the motor speed in a short time. This figure shows the effectiveness of the MFC in rejecting load torque disturbance and maintaining zero steady-state speed error. As a result, the speed loop control performance was good. This result confirms the feasibility of the proposed MFC for speed control of PMa-SynRM.

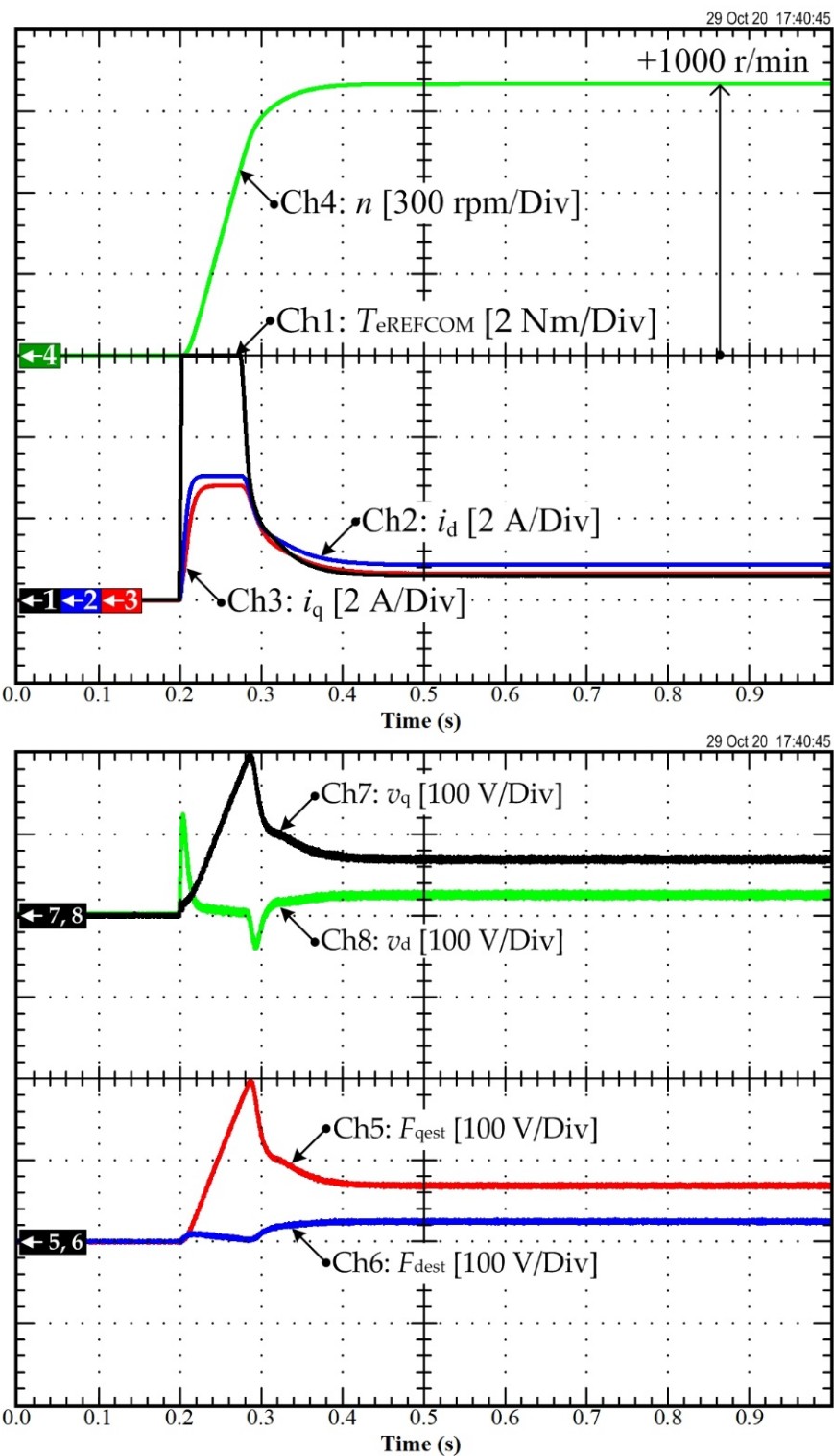

**Figure 12.** Experimental result: Speed acceleration curve based on MFC.

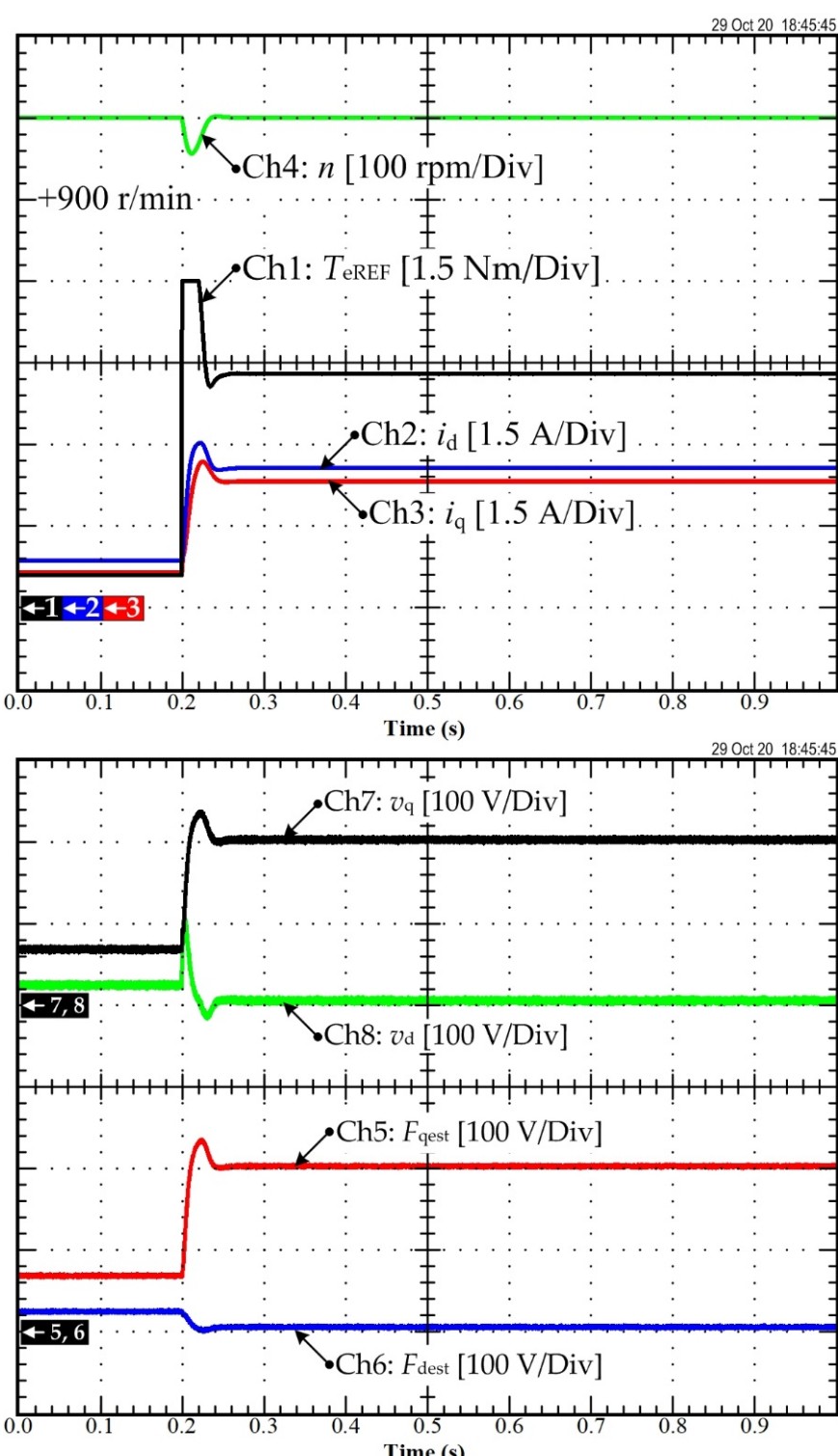

**Figure 13.** Experimental result: Disturbance rejection ability based on MFC.

## 5. Comparison of Traditional FOC with PI Controller, MBC, and Model-Free Control

Traditional FOC based on the PI controller applied to PMa-SynRM was introduced in a previous study [24]. In addition, the MBC based on differential flatness-based control applied to PMa-SynRM was proposed in [13]. Thus, the comparison of the experimental results between the FOC with the PI controller and the MBC (the differential flatness-based control) is expressed below.

Figure 14a shows the current control test of the set-point tracking *d*-axis inner loop of the FOC with the PI controller, and Figure 14b illustrates the current control test of the set-point tracking *d*-axis inner loop of the differential flatness-based control applied to the PMa-SynRM drive system. In Figure 14a,b Ch1 is the current $i_{\text{dCOM}}$, Ch3 is the measured current $i_{\text{d}}$, Ch4 is the measured current $i_{\text{q}}$, and Ch5 is the measured speed *n*. As shown in Figure 14a,b, in a transitory operation, the $i_{\text{d}}$ of the FOC with the PI controller exhibits a small overshoot compared to the differential flatness-based controller, and the $i_{\text{q}}$ of the FOC with PI controller shows oscillations.

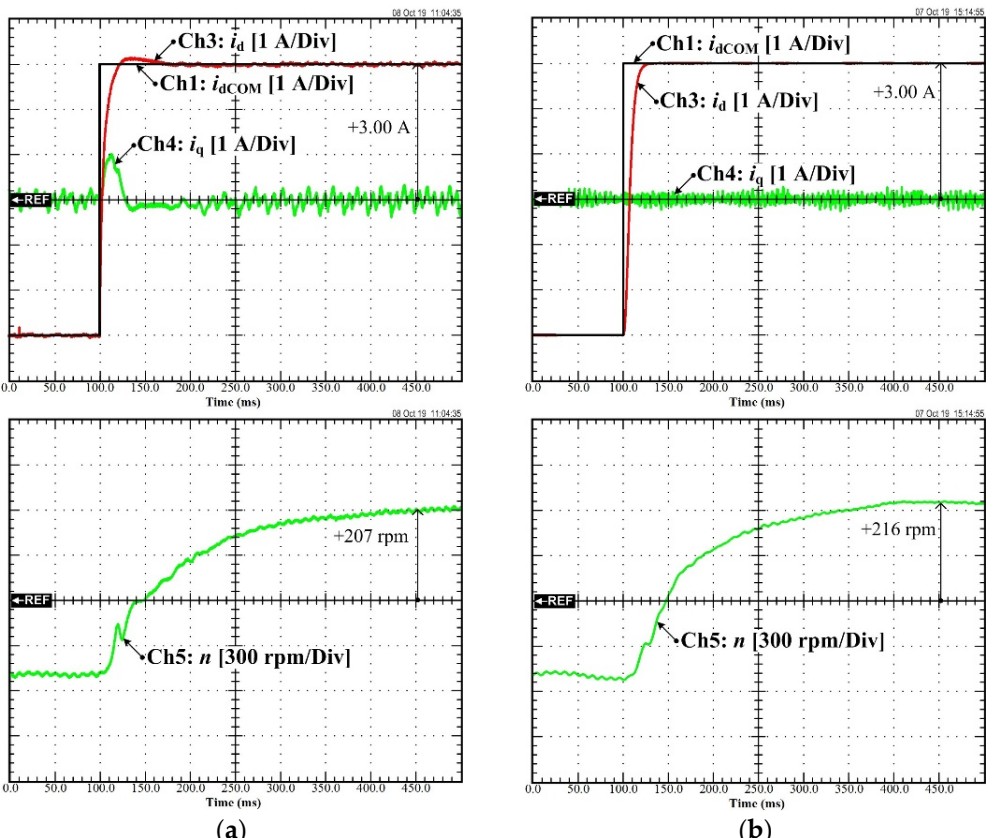

**Figure 14.** Experimental result: Comparison of the set-point tracking between (**a**) the FOC with the PI controller and (**b**) the MBC.

However, the differential flatness-based control was the model-based control (MBC), as mentioned in the introduction. Its performance depends on the system model. More clearly, the control laws of the model-free control and the differential flatness-based control are shown in Figure 15.

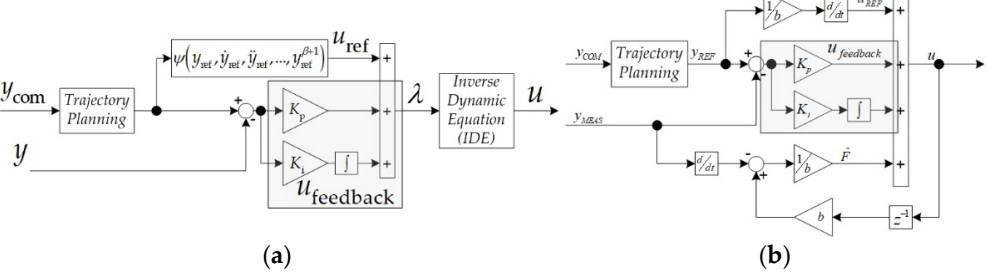

**Figure 15.** The difference between (**a**) the differential flatness-based control law and (**b**) the model-free control law.

The control law of the differential flatness-based control (See Figure 15a) has the inverse dynamic equation, which contains the system models including $R_s$, $L_d$, $L_q$, and $\Psi_m$. In contrast, the control law of the model-free control (See Figure 15b) estimated all the system parameters through the unknown term, $F$.

As a more concise summary, Table 3 shows a comparison of the advantages of traditional FOC+PI, differential flatness-based control, and model-free control.

**Table 3.** Comparison of three different control techniques applied to the PMa-SynRM drive system.

| FOC + PI Controller | Differential Flatness-Based Control | Model-Free Control |
| --- | --- | --- |
| - *More suitable for a linear motor drive system*<br>- *Design controller coefficient using Laplace transform*<br>- *Control performance depending on system parameters* | - *More effective with a nonlinear motor drive system*<br>- *Model-based control system*<br>- *Control performance depending on system parameters*<br>- *Performance enhancement using observer*<br>- *Require more computation resources* | - *Tailored for the control of unknown or partially known systems*<br>- *Partially known parameters (inductance for current control)* |

## 6. Conclusions

In this study, we analyzed the application of an MFC for the current and speed control of motor drives. This novel control approach was applied to PMa-SynRMs for the combined control of the outer speed control loop and inner current control loop. After a brief introduction of the MFC fundamentals, the design approach was comprehensively described, providing a step-by-step procedure. Suggestions for extending the design to different drive controllers were also provided. Simulations and numerous experimental results highlighted the promising features and characteristics of MFC applied to electrical motor drives. Finally, the potential of MFC pointed out in this study should stimulate further exploration and analysis of this type of controller to achieve the expertise required to transfer the results to practical applications.

Interestingly, the proposed MFC provided high performance for the PMa-SynRM drives compared to FOC with the traditional PI controller. Besides, it had a higher dynamic performance than the PMa-SynRM drive using the differential flatness-based control.

In this study, the simulation and the experimental validation were performed by a prototype PMa-SynRM at GREEN Lab, Université de Lorraine. This machine can operate in constant torque and constant power regions if a proper field weakening control is applied. In summary, by applying MFC, the performance of the PMa-SynRM was improved not only in terms of the inner current control loop but also the outer speed control loop. Moreover, the controller coefficients of the proposed MFC are not complicated to define, and a unique design approach can be applied for the PMa-SynRM drive.

**Author Contributions:** Conceptualization, B.N.-M. and N.T.; methodology, S.S., N.P. and P.T.; validation, S.S., N.P. and P.T.; formal analysis, N.B. and P.M.; writing—original draft preparation, P.T.; writing—review and editing, N.B.; visualization, N.B.; supervision, B.N.-M. and N.T. All authors have read and agreed to the published version of the manuscript.

**Funding:** This work was supported in part by the Framework Agreement between the University of Pitesti (Romania) and King Mongkut's University of Technology North Bangkok (Thailand), in part by an International Research Partnership "Electrical Engineering–Thai French Research Center (EE-TFRC)" under the project framework Lorraine Université d'Excellence (LUE) in cooperation with Université de Lorraine and King Mongkut's University of Technology North Bangkok and in part by the National Research Council of Thailand (NRCT) under Senior Research Scholar Program, Grant No. N42A640328, and in part by King Mongkut's University of Technology North Bangkok under Grant no. KMUTNB-64-KNOW-20.

**Institutional Review Board Statement:** Not applicable.

**Informed Consent Statement:** Not applicable.

**Data Availability Statement:** Not applicable.

**Acknowledgments:** The authors would like to express their gratitude to the GREEN laboratory at the University of Lorraine and King Mongkut's University of Technology North Bangkok (KMUTNB) for their constant support in boosting collaborations between France and Thailand. Besides, the authors would like to express their gratitude to the Rajamangala University of Technology Rattanakosin Wang Klai Kangwon Campus as an original agency of the first author (A25/2021).

**Conflicts of Interest:** The authors declare no conflict of interest.

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
