# Peer review of "Design, Modeling, and Model-Free Control of Permanent Magnet-Assisted Synchronous Reluctance Motor for e-Vehicle Applications"

_sustainability, doi:10.3390/su14095423_

Round 1

Reviewer 1 Report

1.The abstract emphasizes that the innovation of this paper is to determine state behavior through a fixed point and transient operation, but the theoretical analysis of fixed point and transient operation is less.

2.The most known and unknown parts of the input are disassembled and combined. The sources of formulas (20) and (21) are not explained. The operator of unknown part is selected to directly replace with the original partial solution, which does not explain whether the selection of estimated quantity will cause an estimated loss compared with the algorithm before the replacement.

3.In this paper, y3 is determined first, and then y1, y2, and whether the estimated part of the unknown part will produce a cumulative error? It was not stated.

4.Figure 5 shows that there is no steady-state error in the process of internal circulation. Should there be some fluctuation error in the actual simulation and final steady state?

5.In the simulation diagram, only the signal tracking in the channel is seen, but the comparison with other algorithms is not seen, and the comparison with other algorithms is only described in the table.

6. When determining the parameters in this paper, the comparison formula is adopted to solve the problem. Is it necessary to demonstrate the convergence of the algorithm during the setting?

Author Response

Dear reviewer,

We are very much thankful to you for your deep and thorough review. We have revised our present research article in light of your useful suggestions and comments. I hope our revision has improved the article to a level of your satisfaction. All the answers to your specific comments/suggestions/queries are listed below.

In order to assist the review process, all corrections made are highlighted in red type in the revised article.

Reviewer 2 Report

  - you must improve the quality of the figures and equations - when commenting on any of the waveforms, do so specifically, adding bullet points to each of the graphs - the selection of references are very recent, with the exception of 2.

- you should make a broader description of the free-model control, even when reference is made to [20] and [21], this would provide more support to your work

 - Are there only 2 differences of this method regarding the FOC and the differential? In table 3, they should also add what this method does not need or require and in the other two methods are mentioned.

Author Response

(The authors gave the same response as above.)

Reviewer 3 Report

In the abstract, it is claimed that the proposed approach outperforms traditional methods. Please indicate in what sense, the proposed approach is better than traditional approaches.

\hat F is calculated in the control signal but it appears in the control signal. This is an algebraic loop which makes the controller un-implementable (see (6) and (8)). Please explain.

In (8), it is mentioned that \hat F is estimation of F. But no adaptation law for F exist in the paper. Please explain how do you do estimation?

Experimental setup needs to be fully explained in the paper.

Please give name to the subfigures. For instance in Fig. 10 you can give it a Fig. 10-a and Fig.10-b for ease of reference.

Author Response

(The authors gave the same response as above.)

Reviewer 4 Report

A model free control scheme for PMa-SRM drive is proposed in this study. The structure of this study is clear, and all the derivations look reasonable. There are 2 suggestions.

  1. In Part 2.1, the fundamentals of model free control should be more extensively explained for readers who are not familiar with this control method before.
  2. Some simulation or experimental results with FOC and other MBCs should be added in Part 4 as counterparts to support the comparison in Part 5.

Author Response

(The authors gave the same response as above.)
